# Aggressive Sampling for Multi-class to Binary Reduction with Applications to Text Classification

**Bikash Joshi**
Univ. Grenoble Alps, LIG
Grenoble, France
bikash.joshi@imag.fr

**Massih-Reza Amini**
Univ. Grenoble Alps, LIG
Grenoble, France
massih-reza.amini@imag.fr

**Ioannis Partalas**
Expedia EWE
Geneva, Switzerland
ipartalas@expedia.com

**Franck Iutzeler**
Univ. Grenoble Alps, LJK
Grenoble, France
franck.iutzeler@imag.fr

**Yury Maximov**
Los Alamos National Laboratory and Skolkovo IST,
USA and Moscow, Russia
yury@lanl.gov

## Abstract

We address the problem of multi-class classification in the case where the number of classes is very large. We propose a double sampling strategy on top of a multi-class to binary reduction strategy, which transforms the original multi-class problem into a binary classification problem over pairs of examples. The aim of the sampling strategy is to overcome the curse of long-tailed class distributions exhibited in majority of large-scale multi-class classification problems and to reduce the number of pairs of examples in the expanded data. We show that this strategy does not alter the consistency of the empirical risk minimization principle defined over the double sample reduction. Experiments are carried out on DMOZ and Wikipedia collections with 10,000 to 100,000 classes where we show the efficiency of the proposed approach in terms of training and prediction time, memory consumption, and predictive performance with respect to state-of-the-art approaches.

## 1 Introduction

Large-scale multi-class or extreme classification involves problems with extremely large number of classes as it appears in text repositories such as Wikipedia, Yahoo! Directory (www.dir.yahoo.com), or Directory Mozilla DMOZ (www.dmoz.org); and it has recently evolved as a popular branch of machine learning with many applications in tagging, recommendation and ranking. The most common and popular baseline in this case is the one-versus-all approach (OVA) [18] where one independent binary classifier is learned per class. Despite its simplicity, this approach suffers from two main limitations; first, it becomes computationally intractable when the number of classes grow large, affecting at the same time the prediction. Second, it suffers from the class imbalance problem by construction.Recently, two main approaches have been studied to cope with these limitations. The first one, broadly divided in tree-based and embedding-based methods, have been proposed with the aim of reducing the effective space of labels in order to control the complexity of the learning problem. Tree-based methods [4, 3, 6, 7, 9, 21, 5, 15] rely on binary tree structures where each leaf corresponds to a class and inference is performed by traversing the tree from top to bottom; a binary classifier being used at each node to determine the child node to develop. These methods have logarithmic time complexity with the drawback that it is a challenging task to find a balanced tree structure which can partition the class labels. Further, even though different heuristics have been developed to address the unbalanced problem, these methods suffer from the drawback that they have to make several decisions prior to reaching a final category, which leads to error propagation and

thus a decrease in accuracy. On the other hand, label embedding approaches [11, 5, 19] first project the label-matrix into a low-dimensional linear subspace and then use an `OVA` classifier. However, the low-rank assumption of the label-matrix is generally transgressed in the extreme multi-class classification setting, and these methods generally lead to high prediction error. The second type of approaches aim at reducing the original multi-class problem into a binary one by first expanding the original training set using a projection of pairs of observations and classes into a low dimensional dyadic space, and then learning a single classifier to separate between pairs constituted with examples and their true classes and pairs constituted with examples with other classes [1, 28, 16]. Although prediction in the new representation space is relatively fast, the construction of the dyadic training observations is generally time consuming and prevails over the training and prediction times.

**Contributions.** In this paper, we propose a scalable multi-class classification method based on an aggressive double sampling of the dyadic output prediction problem. Instead of computing all possible dyadic examples, our proposed approach consists first in drawing a new training set of much smaller size from the original one by oversampling the most small size classes and by sub-sampling the few big size classes in order to avoid the curse of long-tailed class distributions common in the majority of large-scale multi-class classification problems [2]. The second goal is to reduce the number of constructed dyadic examples. Our reduction strategy brings inter-dependency between the pairs containing the same observation and its true class in the original training set. Thus, we derive new generalization bounds using local fractional Rademacher complexity showing that even with a shift in the original class distribution and also the inter-dependency between the pairs of example, the empirical risk minimization principle over the transformation of the sampled training set remains consistent. We validate our approach by conducting a series of experiments on subsets of DMOZ and the Wikipedia collections with up to 100,000 target categories.

## 2    A doubly-sampled multi-class to binary reduction strategy

We address the problem of monolabel multi-class classification defined on joint space $\mathcal{X} \times \mathcal{Y}$ where $\mathcal{X} \subseteq \mathbb{R}^d$ is the *input space* and $\mathcal{Y} = \{1, \ldots, K\} \doteq [K]$ the *output space*, made of $K$ classes. Elements of $\mathcal{X} \times \mathcal{Y}$ are denoted as $\mathbf{x}^y = (x, y)$. Furthermore, we assume the training set $\mathcal{S} = (\mathbf{x}_i^{y_i})_{i=1}^m$ is made of $m$ i.i.d examples/class pairs distributed according to a fixed but unknown probability distribution $\mathcal{D}$, and we consider a class of predictor functions $\mathcal{G} = \{g : \mathcal{X} \times \mathcal{Y} \to \mathbb{R}\}$.

We define the instantaneous loss for predictor $g \in \mathcal{G}$ on example $\mathbf{x}^y$ as:

$$e(g, \mathbf{x}^y) = \frac{1}{K-1} \sum_{y' \in \mathcal{Y} \setminus \{y\}} \mathbb{1}_{g(\mathbf{x}^y) \leq g(\mathbf{x}^{y'})}, \tag{1}$$

where $\mathbb{1}_\pi$ is the indicator function equal to 1 if the predicate $\pi$ is true and 0 otherwise. Compared to the classical multi-class error, $e'(g, \mathbf{x}^y) = \mathbb{1}_{y \neq \operatorname{argmax}_{y' \in \mathcal{Y}} g(\mathbf{x}^{y'})}$, the loss of (1) estimates the average number of classes, given any input data, that get a greater scoring by $g$ than the correct class. The loss (1) is hence a *ranking* criterion, and the multi-class `SVM` of [29] and AdaBoost.MR [24] optimize convex surrogate functions of this loss. It is also used in label ranking [12]. Our objective is to find a function $g \in \mathcal{G}$ with a small expected risk $R(g) = \mathbb{E}_{\mathbf{x}^y \sim \mathcal{D}} [e(g, \mathbf{x}^y)]$, by minimizing the empirical error defined as the average number of training examples $\mathbf{x}_i^{y_i} \in \mathcal{S}$ which, in mean, are scored lower than $\mathbf{x}_i^{y'}$, for $y' \in \mathcal{Y} \setminus \{y_i\}$ :

$$\tilde{R}_m(g, \mathcal{S}) = \frac{1}{m} \sum_{i=1}^m e(g, \mathbf{x}_i^{y_i}) = \frac{1}{m(K-1)} \sum_{i=1}^m \sum_{y' \in \mathcal{Y} \setminus \{y_i\}} \mathbb{1}_{g(\mathbf{x}_i^{y_i}) - g(\mathbf{x}_i^{y'}) \leq 0}. \tag{2}$$

### 2.1    Binary reduction based on dyadic representations of examples and classes

In this work, we consider prediction functions of the form $g = f \circ \phi$, where $\phi : \mathcal{X} \times \mathcal{Y} \to \mathbb{R}^p$ is a projection of the input and the output space into a joint feature space of dimension $p$; and $f \in \mathcal{F} = \{f : \mathbb{R}^p \to \mathbb{R}\}$ is a function that measures the adequacy between an observation $\mathbf{x}$ and a class $y$ using their corresponding representation $\phi(\mathbf{x}^y)$. The projection function $\phi$ is application-dependent and it can either be learned [28], or defined using some heuristics [27, 16].

Further, consider the following dyadic transformation

$$T(\mathcal{S}) = \left( \left\{ \begin{array}{ll} \left(\boldsymbol{z}_j = \left(\phi(\mathbf{x}_i^k), \phi(\mathbf{x}_i^{y_i})\right), \quad \tilde{y}_j = -1\right) & \text{if } k < y_i \\ \left(\boldsymbol{z}_j = \left(\phi(\mathbf{x}_i^{y_i}), \phi(\mathbf{x}_i^k)\right), \quad \tilde{y}_j = +1\right) & \text{elsewhere} \end{array} \right\}_{j \doteq (i-1)(K-1)+k} \right), \tag{3}$$

where $j = (i-1)(K-1)+k$ with $i \in [m], k \in [K-1]$; that expands a $K$-class labeled set $\mathcal{S}$ of size $m$ into a binary labeled set $T(\mathcal{S})$ of size $N = m(K-1)$ (e.g. Figure 1 over a toy problem). With the class of functions

$$\mathcal{H} = \{h : \mathbb{R}^p \times \mathbb{R}^p \to \mathbb{R}; (\phi(\mathbf{x}^y), \phi(\mathbf{x}^{y'})) \mapsto f(\phi(\mathbf{x}^y)) - f(\phi(\mathbf{x}^{y'})), f \in \mathcal{F}\}, \tag{4}$$

the empirical loss (Eq. (2)) can be rewritten as :

$$\tilde{R}_{T(\mathcal{S})}(h) = \frac{1}{N} \sum_{j=1}^{N} \mathbb{1}_{\tilde{y}_j h(\boldsymbol{z}_j) \leq 0}. \tag{5}$$

Hence, the minimization of Eq. (5) over the transformation $T(\mathcal{S})$ of a training set $\mathcal{S}$ defines a binary classification over the pairs of dyadic examples. However, this binary problem takes as examples dependent random variables, as for each original example $\mathbf{x}^y \in \mathcal{S}$, the $K-1$ pairs in $\{(\phi(\mathbf{x}^y), \phi(\mathbf{x}^{y'})); \tilde{y}\} \in T(\mathcal{S})$ all depend on $\mathbf{x}^y$. In [16] this problem is studied by bounding the generalization error associated to (5) using the fractional Rademacher complexity proposed in [25]. In this work, we derive a new generalization bounds based on Local Rademacher Complexities introduced in [22]

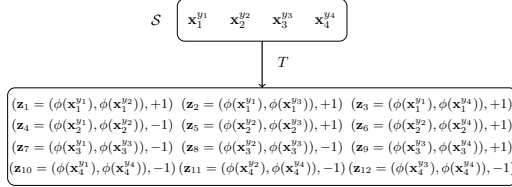

Figure 1: A toy example depicting the transformation $T$ (Eq. (3)) applied to a training set $\mathcal{S}$ of size $m = 4$ and $K = 4$.

that implies second-order (i.e. variance) information inducing faster convergence rates (Theorem 1). Our analysis relies on the notion of graph covering introduced in [14] and defined as :

**Definition 1** (Exact proper fractional cover of $\mathcal{G}$, [14]). *Let $\mathcal{G} = (\mathcal{V}, \mathcal{E})$ be a graph. $\mathcal{C} = \{(\mathcal{C}_k, \omega_k)\}_{k \in [J]}$, for some positive integer $J$, with $\mathcal{C}_k \subseteq \mathcal{V}$ and $\omega_k \in [0,1]$ is an exact proper fractional cover of $\mathcal{G}$, if: i) it is* proper: $\forall k, \mathcal{C}_k$ *is an* independent set, *i.e., there is no connections between vertices in $\mathcal{C}_k$; ii) it is an* exact fractional cover *of $G$: $\forall v \in \mathcal{V}, \sum_{k: v \in \mathcal{C}_k} \omega_k = 1$.*

The weight $W(\mathcal{C})$ of $\mathcal{C}$ is given by: $W(\mathcal{C}) \doteq \sum_{k \in [J]} \omega_k$ and the minimum weight $\chi^*(\mathcal{G}) = \min_{\mathcal{C} \in \mathcal{K}(\mathcal{G})} W(\mathcal{C})$ over the set $\mathcal{K}(\mathcal{G})$ of all exact proper fractional covers of $\mathcal{G}$ is the *fractional chromatic number* of $\mathcal{G}$. From this statement, [14] extended Hoeffding's inequality and proposed large deviation bounds for sums of dependent random variables which was the precursor of new generalisation bounds, including a Talagrand's type inequality for empirical processes in the dependent case presented in [22].

With the classes of functions $\mathcal{G}$ and $\mathcal{H}$ introduced previously, consider the parameterized family $\mathcal{H}_r$ which, for $r > 0$, is defined as:

$$\mathcal{H}_r = \{h : h \in \mathcal{H}, \mathbb{V}[h] \doteq \mathbb{V}_{\boldsymbol{z}, \tilde{y}}[\mathbb{1}_{\tilde{y} h(\boldsymbol{z})}] \leq r\},$$

where $\mathbb{V}$ denotes the variance.

The fractional Rademacher complexity introduced in [25] entails our analysis :

$$\mathfrak{R}_{T(\mathcal{S})}(\mathcal{H}) \doteq \frac{2}{N} \mathbb{E}_\xi \sum_{k \in [K-1]} \omega_k \mathbb{E}_{\mathcal{C}_k} \sup_{h \in \mathcal{H}} \sum_{\substack{\alpha \in \mathcal{C}_k \\ \boldsymbol{z}_\alpha \in T(\mathcal{S})}} \xi_\alpha h(\boldsymbol{z}_\alpha),$$

with $(\xi_i)_{i=1}^N$ a sequence of independent Rademacher variables verifying $\mathbb{P}(\xi_n = 1) = \mathbb{P}(\xi_n = -1) = \frac{1}{2}$. If other is not specified explicitly we assume below all $\omega_k = 1$. Our first result that bounds the generalization error of a function $h \in \mathcal{H}$; $R(h) = \mathbb{E}_{T(\mathcal{S})}[\tilde{R}_{T(\mathcal{S})}(h)]$, with respect to its empirical error $\tilde{R}_{T(\mathcal{S})}(h)$ over a transformed training set, $T(\mathcal{S})$, and the fractional Rademacher complexity, $\mathfrak{R}_{T(\mathcal{S})}(\mathcal{H})$, is stated below.

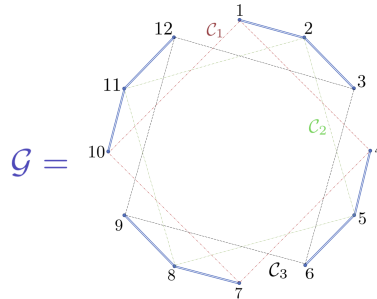

Figure 2: The dependency graph $\mathcal{G} = \{1, \ldots, 12\}$ corresponding to the toy problem of Figure 1, where dependent nodes are connected with vertices in blue double-line. The exact proper fractional cover $\mathcal{C}_1$, $\mathcal{C}_2$ and $\mathcal{C}_3$ is shown in dashed. The fractional chromatic number is in this case $\chi^*(\mathcal{G}) = K - 1 = 3$.

**Theorem 1.** *Let $\mathcal{S} = (\mathbf{x}_i^{y_i})_{i=1}^m \in (\mathcal{X} \times \mathcal{Y})^m$ be a dataset of $m$ examples drawn i.i.d. according to a probability distribution $\mathcal{D}$ over $\mathcal{X} \times \mathcal{Y}$ and $T(\mathcal{S}) = ((\mathbf{z}_i, \tilde{y}_i))_{i=1}^N$ the transformed set obtained as in Eq. (3). Then for any $1 > \delta > 0$ and $0/1$ loss $\ell : \{-1, +1\} \times \mathbb{R} \rightarrow [0, 1]$, with probability at least $(1 - \delta)$ the following generalization bound holds for all $h \in \mathcal{H}_r$ :*

$$R(h) \leq \tilde{R}_{T(\mathcal{S})}(h) + \mathfrak{R}_{T(\mathcal{S})}(\ell \circ \mathcal{H}_r) + \frac{5}{2}\left(\sqrt{\mathfrak{R}_{T(\mathcal{S})}(\ell \circ \mathcal{H}_r)} + \sqrt{\frac{r}{2}}\right)\sqrt{\frac{\log\frac{1}{\delta}}{m}} + \frac{25}{48}\frac{\log\frac{1}{\delta}}{m}.$$

The proof is provided in the supplementary material, and it relies on the idea of splitting up the sum (5) into several parts, each part being a sum of independent variables.

## 2.2 Aggressive Double Sampling

Even-though the previous multi-class to binary transformation $T$ with a proper projection function $\phi$ allows to redefine the learning problem in a dyadic feature space of dimension $p \ll d$, the increased number of examples can lead to a large computational overhead. In order to cope with this problem, we propose a $(\pi, \kappa)$-*double subsampling* of $T(\mathcal{S})$, which first aims to balance the presence of classes by constructing a new training set $\mathcal{S}_\pi$ from $\mathcal{S}$ with probabilities $\pi = (\pi_k)_{k=1}^K$.

The idea here is to overcome the curse of long-tailed class distributions exhibited in majority of large-scale multiclass classification problems [2] by oversampling the most small size classes and by subsampling the few big size classes in $\mathcal{S}$. The hyperparameters $\pi$ are formally defined as $\forall k, \pi_k = P(\mathbf{x}^y \in \mathcal{S}_\pi | \mathbf{x}^y \in \mathcal{S})$. In practice we set them inversely proportional to the size of each class in the original training set; $\forall k, \pi_k \propto 1/\mu_k$ where $\mu_k$ is the proportion of class $k$ in $\mathcal{S}$. The second aim is to reduce the number of dyadic examples controlled by the

---

**Algorithm:** $(\pi, \kappa)$-DS

**Input:** Labeled training set $\mathcal{S} = (\mathbf{x}_i^{y_i})_{i=1}^m$
**initialization:** $\mathcal{S}_\pi \leftarrow \emptyset$;
$T_\kappa(\mathcal{S}_\pi) \leftarrow \emptyset$ ;
**for** $k = 1..K$ **do**
     Draw randomly a set $\mathcal{S}_{\pi_k}$ of examples of class $k$ from $\mathcal{S}$ with
       probability $\pi_k$;
     $\mathcal{S}_\pi \leftarrow \mathcal{S}_\pi \cup \mathcal{S}_{\pi_k}$;
**forall** $\mathbf{x}^y \in \mathcal{S}_\pi$ **do**
     Draw uniformly a set $\mathcal{Y}_{\mathbf{x}^y}$ of $\kappa$ classes from $\mathcal{Y}\backslash\{y\}$   $\triangleright \kappa \ll K$;
     **forall** $k \in \mathcal{Y}_{\mathbf{x}^y}$ **do**
         **if** $k < y$ **then**
             $T_\kappa(\mathcal{S}_\pi) \leftarrow T_\kappa(\mathcal{S}_\pi) \cup (\mathbf{z} = (\phi(\mathbf{x}^k), \phi(\mathbf{x}^y)), \; \tilde{y} = -1)$;
         **else**
             $T_\kappa(\mathcal{S}_\pi) \leftarrow T_\kappa(\mathcal{S}_\pi) \cup (\mathbf{z} = (\phi(\mathbf{x}^y), \phi(\mathbf{x}^k)), \; \tilde{y} = +1)$;

**return** $T_\kappa(\mathcal{S}_\pi)$

---

hyperparameter $\kappa$. The pseudo-code of this *aggressive double sampling* procedure, referred to as $(\pi, \kappa)$-DS, is depicted above and it is composed of two main steps.

1. For each class $k \in \{1, \ldots, K\}$, draw randomly a set $\mathcal{S}_{\pi_k}$ of examples from $\mathcal{S}$ of that class with probability $\pi_k$, and let $\mathcal{S}_\pi = \bigcup_{k=1}^K \mathcal{S}_{\pi_k}$;

2. For each example $\mathbf{x}^y$ in $\mathcal{S}_\pi$, draw uniformly $\kappa$ adversarial classes in $\mathcal{Y}\backslash\{y\}$.

After this double sampling, we apply the transformation $T$ defined as in Eq. (3), leading to a set $T_\kappa(S_\pi)$ of size $\kappa|\mathcal{S}_\pi| \ll N$.

In Section 3, we will show that this procedure practically leads to dramatic improvements in terms of memory consumption, computational complexity, and a higher multi-class prediction accuracy when the number of classes is very large. The empirical loss over the transformation of the new subsampled training set $\mathcal{S}_\pi$ of size $M$, outputted by the $(\pi, \kappa)$-DS algorithm is :

$$\tilde{R}_{T_\kappa(\mathcal{S}_\pi)}(h) = \frac{1}{\kappa M}\sum_{(\tilde{y}_\alpha, \mathbf{z}_\alpha) \in T_\kappa(\mathcal{S}_\pi)} \mathbb{1}_{\tilde{y}_\alpha h(z_\alpha) \leq 0} = \frac{1}{\kappa M}\sum_{\mathbf{x}^y \in \mathcal{S}_\pi}\sum_{y' \in \mathcal{Y}_{\mathbf{x}^y}} \mathbb{1}_{g(\mathbf{x}^y) - g(\mathbf{x}^{y'}) \leq 0}, \quad (6)$$

which is essentially the same empirical risk as the one defined in Eq. (2) but taken with respect to the training set outputted by the $(\pi, \kappa)$-DS algorithm. Our main result is the following theorem which bounds the generalization error of a function $h \in \mathcal{H}$ learned by minimizing $\tilde{R}_{T_\kappa(\mathcal{S}_\pi)}$.

**Theorem 2.** *Let $\mathcal{S} = (\mathbf{x}_i^{y_i})_{i=1}^m \in (\mathcal{X} \times \mathcal{Y})^m$ be a training set of size $m$ i.i.d. according to a probability distribution $\mathcal{D}$ over $\mathcal{X} \times \mathcal{Y}$, and $T(\mathcal{S}) = ((\mathbf{z}_i, \tilde{y}_i))_{i=1}^N$ the transformed set obtained with the transformation function $T$ defined as in Eq. (3). Let $\mathcal{S}_\pi \subseteq \mathcal{S}$, $|\mathcal{S}_\pi| = M$, be a training set outputted by the algorithm $(\pi, \kappa)$-DS and $T(\mathcal{S}_\pi) \subseteq T(\mathcal{S})$ its corresponding transformation. Then for any $1 > \delta > 0$ with probability at least $(1 - \delta)$ the following risk bound holds for all $h \in \mathcal{H}$ :*

$$R(h) \le \alpha \tilde{R}_{T_\kappa(\mathcal{S}_\pi)}(h) + \alpha \mathfrak{R}_{T_\kappa(\mathcal{S}_\pi)}(\ell \circ \mathcal{H}) + \alpha \sqrt{\frac{(K-1)\log\frac{2}{\delta}}{2M\kappa}} + \sqrt{\frac{2\alpha \log\frac{4K}{\delta}}{\beta(m-1)}} + \frac{7\beta \log\frac{4K}{\delta}}{3(m-1)}.$$

*Furthermore, for all functions in the class $\mathcal{H}_r$, we have the following generalization bound that holds with probability at least $(1 - \delta)$ :*

$$R(h) \le \alpha \tilde{R}_{T_\kappa(\mathcal{S}_\pi)}(h) + \alpha \mathfrak{R}_{T_\kappa(\mathcal{S}_\pi)}(\ell \circ \mathcal{H}_r) + \sqrt{\frac{2\alpha \log\frac{4K}{\delta}}{\beta(m-1)}} + \frac{7\beta \log\frac{4K}{\delta}}{3(m-1)}$$

$$+ \frac{5\alpha}{2}\left(\sqrt{\mathfrak{R}_{T_\kappa(\mathcal{S}_\pi)}(\ell \circ \mathcal{H}_r)} + \sqrt{\frac{r}{2}}\right)\sqrt{\frac{(K-1)\log\frac{2}{\delta}}{M\kappa}} + \frac{25\alpha}{48}\frac{\log\frac{2}{\delta}}{M},$$

*where $\ell : \{-1, +1\} \times \mathbb{R} \to [0, 1]$ 0/1 is an instantaneous loss, and $\alpha = \max_{y: 1 \le y \le K} \eta_y / \pi_y$, $\beta = \max_{y: 1 \le y \le K} 1/\pi_y$ and $\eta_y > 0$ is the proportion of class $y$ in $\mathcal{S}$.*

The proof is provided in the supplementary material. This theorem hence paves the way for the consistency of the empirical risk minimization principle [26, Th. 2.1, p. 38] defined over the double sample reduction strategy we propose.

## 2.3 Prediction with Candidate Selection

The prediction is carried out in the dyadic feature space, by first considering the pairs constituted by a test observation and all the classes, and then choosing the class that leads to the highest score by the learned classifier. In the large scale scenario, computing the feature representations for all classes may require a huge amount of time. To overcome this problem we sample over classes by choosing just those that are the nearest to a test example, based on its distance with class centroids. Here we propose to consider class centroids as the average of vectors

---

**Algorithm:** Prediction with Candidate Selection Algorithm

**Input:** Unlabeled test set $\mathcal{T}$;
Learned function $f^* : \mathbb{R}^p \to \mathbb{R}$;
**initialization:** $\Omega \leftarrow \emptyset$;
**forall $\mathbf{x} \in \mathcal{T}$ do**
    Select $\mathcal{Y}_\mathbf{x} \subseteq \mathcal{Y}$ candidate set of $q$ nearest-centroid classes;
    $\Omega \leftarrow \Omega \cup \text{argmax}_{k \in \mathcal{Y}_\mathbf{x}} f^*(\phi(\mathbf{x}^k))$ ;
**return** *predicted classes* $\Omega$

---

within that class. Note that class centroids are computed once in the preliminary projection of training examples and classes in the dyadic feature space and thus represent no additional computation at this stage. The algorithm above presents the pseudocode of this candidate based selection strategy [1].

# 3 Experiments

In this section, we provide an empirical evaluation of the proposed reduction approach with the $(\pi, \kappa)$-DS sampling strategy for large-scale multi-class classification of document collections. First, we present the mapping $\phi : \mathcal{X} \times \mathcal{Y} \to \mathbb{R}^p$. Then, we provide a statistical and computational comparison of our method with state-of-the-art large-scale approaches on popular datasets.

## 3.1 a Joint example/class representation for text classification

The particularity of text classification is that documents are represented in a vector space induced by the vocabulary of the corresponding collection [23]. Hence each class can be considered as a mega-document, constituted by the concatenation of all of the documents in the training set belonging to it,

| Features in the joint example/class representation representation $\phi(\mathbf{x}^y)$. |
|---|

**1.** $\sum_{t \in y \cap \mathbf{x}} \log(1 + y_t)$   **2.** $\sum_{t \in y \cap \mathbf{x}} \log\left(1 + \frac{l_{\mathcal{S}}}{F_t}\right)$   **3.** $\sum_{t \in y \cap \mathbf{x}} I_t$

**4.** $\sum_{t \in y \cap \mathbf{x}} \frac{y_t}{|y|} . I_t$   **5.** $\sum_{t \in y \cap \mathbf{x}} \log\left(1 + \frac{y_t}{|y|}\right)$   **6.** $\sum_{t \in y \cap \mathbf{x}} \log\left(1 + \frac{y_t}{|y|} . I_t\right)$

**7.** $\sum_{t \in y \cap \mathbf{x}} \log\left(1 + \frac{y_t}{|y|} . \frac{l_{\mathcal{S}}}{F_t}\right)$   **8.** $\sum_{t \in y \cap \mathbf{x}} 1$   **9.** $d(\mathbf{x}^y, \texttt{centroid}(y))$

**10.** $\mathrm{BM25} = \sum_{t \in y \cap x} I_t . \frac{2 \times y_t}{y_t + (0.25 + 0.75 \cdot \texttt{len}(y)/\texttt{avg}(\texttt{len}(y)))}$

Table 1: Joint example/class representation for text classification, where $t \in y \cap \mathbf{x}$ are terms that are present in both the class $y$'s mega-document and document $\mathbf{x}$. $\mathcal{V}$ represents the set of distinct terms within $\mathcal{S}$, and $\mathbf{x}_t$ is the frequency of term $t$ in $\mathbf{x}$, $y_t = \sum_{\mathbf{x} \in y} \mathbf{x}_t$, $|y| = \sum_{t \in \mathcal{V}} y_t$, $F_t = \sum_{\mathbf{x} \in \mathcal{S}} \mathbf{x}_t$, $l_{\mathcal{S}} = \sum_{t \in \mathcal{V}} \mathcal{S}_t$. Finally, $I_t$ is the inverse document frequency of term $t$, $len(y)$ is number of terms of documents in class $y$, and $avg(len(y))$ is the average of document lengths for all the classes.

and simple feature mapping of examples and classes can be defined over their common words. Here we used $p = 10$ features inspired from learning to rank [17] by resembling a class and a document to respectively a document and a query (Table 1). All features except feature 9, that is the distance of an example $x$ to the centroid of all examples of a particular class $y$, are classical. In addition to its predictive interest, the latter is also used in prediction for performing candidate preselection. Note that for other large-scale multi-class classification applications like recommendation with extremely large number of offer categories or image classification, a same kind of mapping can either be learned or defined using their characteristics [27, 28].

## 3.2 Experimental Setup

**Datasets.** We evaluate the proposed method using popular datasets from the Large Scale Hierarchical Text Classification challenge (LSHTC) 1 and 2 [20]. These datasets are provided in a pre-processed format using stop-word removal and stemming. Various characteristics of these datesets including the statistics of train, test and heldout are listed in Table 2. Since, the datasets used in LSHTC2 challenge were in multi-label format, we converted them to multi-class format by replicating the instances belonging to different class labels. Also, for the largest dataset (WIKI-large) used in LSHTC2 challenge, we used samples with 50,000 and 100,000 classes. The smaller dataset of LSHTC2 challenge is named as WIKI-Small, whereas the two 50K and 100K samples of large dataset are named as WIKI-50K and WIKI-100K in our result section.

| Datasets | # of classes, $K$ | Train Size | Test Size | Heldout Size | Dimension, $d$ |
|---|---|---|---|---|---|
| LSHTC1 | 12294 | 126871 | 31718 | 5000 | 409774 |
| DMOZ | 27875 | 381149 | 95288 | 34506 | 594158 |
| WIKI-Small | 36504 | 796617 | 199155 | 5000 | 380078 |
| WIKI-50K | 50000 | 1102754 | 276939 | 5000 | 951558 |
| WIKI-100K | 100000 | 2195530 | 550133 | 5000 | 1271710 |

Table 2: Characteristics of the datasets used in our experiments

**Baselines.** We compare the proposed approach,[2] denoted as the sampling strategy by $(\pi, \kappa)$-DS, with popular baselines listed below:

- OVA: LibLinear [10] implementation of one-vs-all SVM.

- M-SVM: LibLinear implementation of multi-class SVM proposed in [8].

- RecallTree [9]: A recent tree based multi-class classifier implemented in Vowpal Wabbit.

| Data | | OVA | M-SVM | RecallTree | FastXML | PfastReXML | PD-Sparse | $(\pi, \kappa)$-DS |
|---|---|---|---|---|---|---|---|---|
| **LSHTC1** | train time | 23056s | 48313s | 701s | 8564s | 3912s | 5105s | 321s |
| m = 163589 | predict time | 328s | 314s | 21s | 339s | 164s | 67s | 544s |
| d = 409774 | total memory | 40.3G | 40.3G | 122M | 470M | 471M | 10.5G | 2G |
| K = 12294 | Accuracy | 44.1% | 36.4% | 18.1% | 39.3% | 39.8% | 45.7% | 37.4% |
| | MaF$_1$ | 27.4% | 18.8% | 3.8% | 21.3% | 22.4% | 27.7% | 26.5% |
| **DMOZ** | train time | 180361s | 212356s | 2212s | 14334s | 15492s | 63286s | 1060s |
| m = 510943 | predict time | 2797s | 3981s | 47s | 424s | 505s | 482s | 2122s |
| d = 594158 | total memory | 131.9G | 131.9G | 256M | 1339M | 1242M | 28.1G | 5.3G |
| K = 27875 | Accuracy | 37.7% | 32.2% | 16.9% | 33.4% | 33.7% | 40.8% | 27.8% |
| | MaF$_1$ | 22.2% | 14.3% | 1.75% | 15.1% | 15.9% | 22.7% | 20.5% |
| **WIKI-Small** | train time | 212438s | >4d | 1610s | 10646s | 21702s | 16309s | 1290s |
| m = 1000772 | predict time | 2270s | NA | 24s | 453s | 871s | 382s | 2577s |
| d = 380078 | total memory | 109.1G | 109.1G | 178M | 949M | 947M | 12.4G | 3.6G |
| K = 36504 | Accuracy | 15.6% | NA | 7.9% | 11.1% | 12.1% | 15.6% | 21.5% |
| | MaF$_1$ | 8.8 % | NA | <1% | 4.6% | 5.63% | 9.91% | 13.3% |
| **WIKI-50K** | train time | NA | NA | 4188s | 30459s | 48739s | 41091s | 3723s |
| m = 1384693 | predict time | NA | NA | 45s | 1110s | 2461s | 790s | 4083s |
| d = 951558 | total memory | 330G | 330G | 226M | 1327M | 1781M | 35G | 5G |
| K = 50000 | Accuracy | NA | NA | 17.9% | 25.8% | 27.3% | 33.8% | 33.4% |
| | MaF$_1$ | NA | NA | 5.5% | 14.6% | 16.3% | 23.4% | 24.5% |
| **WIKI-100K** | train time | NA | NA | 8593s | 42359s | 73371s | 155633s | 9264s |
| m = 2750663 | predict time | NA | NA | 90s | 1687s | 3210s | 3121s | 20324s |
| d = 1271710 | total memory | 1017G | 1017G | 370M | 2622M | 2834M | 40.3G | 9.8G |
| K = 100000 | Accuracy | NA | NA | 8.4% | 15% | 16.1% | 22.2% | 25% |
| | MaF$_1$ | NA | NA | 1.4% | 8% | 9% | 15.1% | 17.8% |

Table 3: Comparison of the result of various baselines in terms of time, memory, accuracy, and macro F1-measure

- FastXML [21]: An extreme multi-class classification method which performs partitioning in the feature space for faster prediction.
- PfastReXML [13]: Tree ensemble based extreme classifier for multi-class and multilabel problems.
- PD-Sparse [30]: A recent approach which uses multi-class loss with $\ell_1$-regularization.

Referring to the work [30], we did not consider other recent methods SLEEC [5] and LEML [31] in our experiments, since they have been shown to be consistently outperformed by the above mentioned state-of-the-art approaches.

**Platform and Parameters.** In all of our experiments, we used a machine with an Intel Xeon 2.60GHz processor with 256 GB of RAM. Each of these methods require tuning of various hyper-parameters that influence their performance. For each methods, we tuned the hyperparameters over a heldout set and used the combination which gave best predictive performance. The list of used hyperparameters for the results we obtained are reported in the supplementary material (Appendix B).

**Evaluation Measures.** Different approaches are evaluated over the test sets using accuracy and the macro F1 measure (MaF$_1$), which is the harmonic average of macro precision and macro recall; higher MaF$_1$ thus corresponds to better performance. As opposed to accuracy, macro F1 measure is not affected by the class imbalance problem inherent to multi-class classification, and is commonly used as a robust measure for comparing predictive performance of classification methods.

# 4 Results

The parameters of the datasets along with the results for compared methods are shown in Table 3. The results are provided in terms of train and predict times, total memory usage, and predictive performance measured with accuracy and macro F1-measure (MaF$_1$). For better visualization and comparison, we plot the same results as bar plots in Fig. 3 keeping only the best five methods while comparing the total runtime and memory usage. First, we observe that the tree based approaches (FastXML, PfastReXML and RecallTree) have worse predictive performance compared to the other methods. This is due to the fact that the prediction error made at the top-level of the tree cannot be corrected at lower levels, also known as cascading effect. Even though they have lower runtime and memory usage, they suffer from this side effect.

For large scale collections (**WIKI-Small**, **WIKI-50K** and **WIKI-100K**), the solvers with competitive predictive performance are OVA, M-SVM, PD-Sparse and $(\pi, \kappa)$-DS. However, standard OVA and

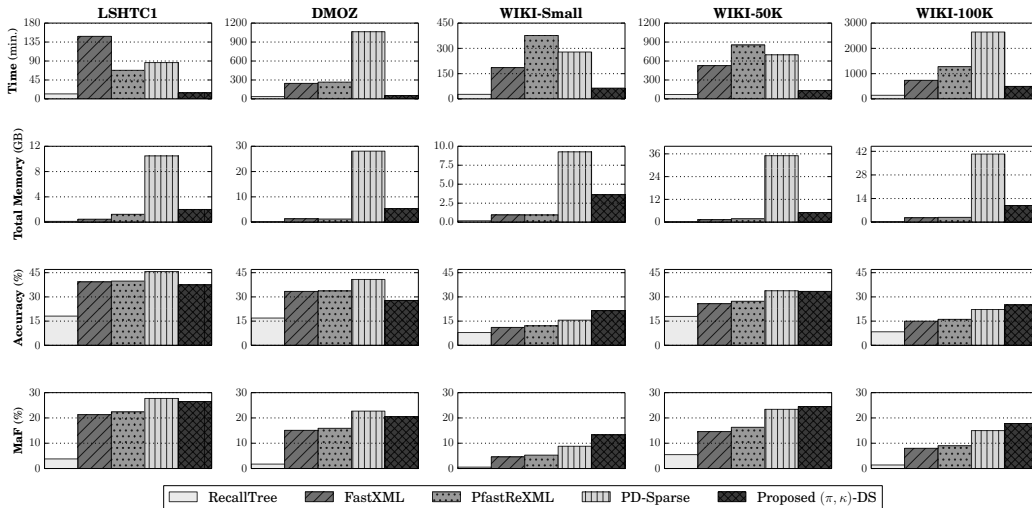

Figure 3: Comparisons in Total (Train and Test) Time (min.), Total Memory usage (GB), and $\text{MaF}_1$ of the five best performing methods on **LSHTC1**, **DMOZ**, **WIKI-Small**, **WIKI-50K** and **WIKI-100K**.

`M-SVM` have a complexity that grows linearly with $K$ thus the total runtime and memory usage explodes on those datasets, making them impossible. For instance, on Wiki large dataset sample of 100K classes, the memory consumption of both approaches exceeds the Terabyte and they take several days to complete the training. Furthermore, on this data set and the second largest Wikipedia collection (**WIKI-50K** and **WIKI-100K**) the proposed approach is highly competitive in terms of Time, Total Memory and both performance measures comparatively to all the other approaches. These results suggest that the method least affected by long-tailed class distributions is $(\pi, \kappa)$-`DS`, mainly because of two reasons: first, the sampling tends to make the training set balanced and second, the reduced binary dataset contains similar number of positive and negative examples. Hence, for the proposed approach, there is an improvement in both accuracy and $\text{MaF}_1$ measures. The recent `PD-Sparse` method also enjoys a competitive predictive performance but it requires to store intermediary weight vectors during optimization which prevents it from scaling well. The `PD-Sparse` solver provides an option for hashing leading to fewer memory usage during training which we used in the experiments; however, the memory usage is still significantly high for large datasets and at the same time this option slows down the training process considerably. In overall, among the methods with competitive predictive performance, $(\pi, \kappa)$-`DS` seems to present the best runtime and memory usage; its runtime is even competitive with most of tree-based methods, leading it to provide the best compromise among the compared methods over the three time, memory and performance measures.

## 5 Conclusion

We presented a new method for reducing a multiclass classification problem to binary classification. We employ similarity based feature representation for class and examples and a double sampling stochastic scheme for the reduction process. Even-though the sampling scheme shifts the distribution of classes and that the reduction of the original problem to a binary classification problem brings inter-dependency between the dyadic examples; we provide generalization error bounds suggesting that the Empirical Risk Minimization principle over the transformation of the sampled training set still remains consistent. Furthermore, the characteristics of the algorithm contribute for its excellent performance in terms of memory usage and total runtime and make the proposed approach highly suitable for large class scenario.

#### Acknowledgments

This work has been partially supported by the LabEx PERSYVAL-Lab (ANR-11-LABX-0025-01) funded by the French program *Investissement d'avenir*, and by the U.S. Department of Energy's Office of Electricity as part of the DOE Grid Modernization Initiative.

## Footnotes

[1]The number of classes pre-selected can be tuned to offer a prediction time/accuracy tradeoff if the prediction time is more critical.

[2]Source code and datasets can be found in the following repository https://github.com/bikash617/Aggressive-Sampling-for-Multi-class-to-BinaryReduction

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
