[Reviews · NeurIPS 2017]

Reviewer 1



This paper presents a double sampling strategy to improve the multiclass classification approach in [16]. The authors present both theoretical and empirical analysis of their proposed approach. Experiments conducted on text data set shows the proposed approach can handle multi-class classification problem with a large number of classes. Pros: - The authors conduct comprehensive experimental comparisons with several baseline methods. Results show that the proposed approach achieves high performance with shorter training time on several datasets. - The extreme classification problem is an important direction of research and have many applications in practice. Cons: - The paper is mostly based on [16]. Despite there are two main improvements: 1) proposed the double sampling strategy and 2) a new generalization bounds based on local Rademacher Complexities, the novelty is relatively thin. - Despite the proposed approach shorten the training time, it still requires long prediction time compared with other approaches. It is arguably the prediction time is more important in practice. Comments: - It is unclear how to interoperate the generalization bound presented in the paper. How the generation bound compared with other approaches? - Some methods presented in the experiments are batch learning algorithms and some are online learning algorithms. Therefore, the memory usages are very different.

Reviewer 2



Summary: This paper proposes a new reduction from multi-class classification to binary classification that is especially suitable when the number of classes is very large. They consider a hypothesis that map (input,class) pairs to scores, and the underlying loss function counts the fraction of the wrong classes that are scored higher than the true class. More specifically, they suppose they have a feature transformation phi that maps (input,class) pairs to a p-dimensional feature space, and they learn a mapping from R^p to scores. Their reduction extends the work of Joshi et al. (2015) which, for each data point (x,y), creates K-1 transformed points where each transformed point intuitively corresponds to the comparison of label y with some incorrect label y'. Given that the transformed dataset contains correlated training examples, many standard generalization bounds cannot be applied. The first contribution of this paper is an improved generalization analysis over Joshi et al. (2015) using new bounds on the fractional Rademacher complexity in this setting. The second main contribution is showing that it is possible to employ sampling techniques during the reduction to balance the class distributions and to decrease the training set size. In particular, they sample each class (with replacement) in order to ensure that each class has roughly equal representation in the transformed training data, and rather than including transformed points for all K-1 possible class comparisons, they use only a random sample of classes. They show that these two sampling procedures do not drastically affect their generalization bounds. Finally, they show that this new procedure performs very competitively against several strong baselines. Comments: This paper presents an interesting reduction from multi-class classification with many classes to binary classification and corresponding generalization bounds for dealing with the fact that the reduced dataset contains correlated data. The main novelty of the reduction presented in the paper is the sampling technique for rebalancing the class distributions and for reducing the size of the transformed dataset, while the basic reduction appeared already in an earlier paper by Joshi etl al. The empirical comparison to existing algorithms are very strong. One thing that is missing from the paper is a theoretical analysis of the impact of the prediction algorithm given in section 2.3, where comparisons are only performed for the k classes whose centroids are closest to a given input point. I would have also appreciated some additional discussion of the sample complexity bounds, possibly comparing them to the generalization bounds obtained for the non-transformed multi-class problems.

Reviewer 3



This paper presents a learning reduction for extreme classification, multiclass classification where the output space ranges up 100k different classes. Many approaches to extreme classification rely on inferring a tree structure among the output labels, as in the hierarchical softmax, so that only a logarithmic number of binary predictions need to be made to infer the output class, or a label embedding approach that can be learned efficiently through least-squares or sampling. The authors note the problems with these approaches: the inferred latent tree may not be optimal and lead to cascading errors, and the label embedding approach may lead to prediction errors when the true label matrix is not low-rank. An alternative approach is to reduce extreme classification to pairwise binary classification. The authors operate in the second framework and present a scalable sampling based method with theoretical guarantees as to its consistency with the original multiclass ERM formulation. Experiments on 5 text datasets comparing a number of competing approaches (including standard one-vs-all classification when that is computationally feasible) show that the method is generally superior in terms of accuracy and F1 once the number of classes goes above 30k, and trains faster while using far less memory, although suffers somewhat in terms of prediction speed. The Aggressive Double Sampling reduction is controlled by two parameters, one of which controls the frequency of sampling each class, which is inversely proportional to the empirical class frequency in order to avoid entirely missing classes in the long tail. The second parameter sets the number of adversarial examples to draw uniformly. This procedure is repeated to train the final dyadic classifier. At prediction time, inference is complicated by the fact that generating pairwise features with all possible classes is intractable. A heuristic is used to identify candidate classes by making an input space centroid representation of each class as the average of vectors in the class, with prediction being carried out by a series of pairwise classifications with only those candidates. Theoretical analysis of the reduction is complicated by the fact that the reduction from multi-class to dyadic binary classification changes a sum of independent random variables into a sum of random variables with a particular dependence structure among the dyadic examples. The other complication is caused by the oversampling of rare classes, shifting the empirical distribution. The proof involves splitting the sum of dependent examples into several sums of independent variables based on the graph structure induced by the example construction, and using the concentration equalities for partially dependent variables introduced by Janson. Although the finite-sample risk bound is biased by the ratios between the true class probabilities and the oversampled ones, these biases decrease linearly with the size of the sampled and re-sampled training sets. The analysis appears sound but I have not verified the proof in detail. Overall, this paper presents a strong method for extreme classification, with small memory and computational requirements and superior performance on especially the largest datasets. The binary reduction algorithm appears theoretically sound, interesting, and useful to practitioners.